# The Erythrocyte Sedimentation Rate (ESR) in Veterinary Medicine: A Focused Review in Dogs and Cats

**DOI:** 10.3390/ani15020246

**Published:** 2025-01-16

**Authors:** Daniela Diamanti, Carolina Pieroni, Maria Grazia Pennisi, Veronica Marchetti, Eleonora Gori, Saverio Paltrinieri, George Lubas

**Affiliations:** 1DIESSE Diagnostica Senese SpA, 53035 Monteriggioni, Italy; danieladiamanti@diesse.it (D.D.); carolinapieroni@diesse.it (C.P.); 2“I Periodeuti” ASC, 89122 Reggio Calabria, Italy; mariagrazia.pennisi@unime.it; 3Department of Veterinary Sciences, University of Pisa, 56121 Pisa, Italy; veronica.marchetti@unipi.it (V.M.); eleonora.gori@unipi.it (E.G.); 4Department of Veterinary Medicine and Animal Sciences, University of Milan, 26900 Lodi, Italy; saverio.paltrinieri@unimi.it; 5Clinica Veterinaria Colombo, VetPartners Italia, V.le Colombo 153, 55041 Camaiore, Italy

**Keywords:** erythrocyte sedimentation rate, ESR, modified Westergren ESR, dog, cat, reference intervals, diseases

## Abstract

The erythrocyte sedimentation rate (ESR) is a diagnostic test that measures the rate at which red blood cells settle in anticoagulated blood. The ESR is directly linked to an increase in acute response proteins such as fibrinogen, immunoglobulin, and macroglobulin which promote erythrocyte aggregation. The ESR is, thus, useful for detecting inflammation derived from infectious diseases, urinary or orthopedic disorders, or miscellaneous diseases. The current gold standard is still Westergren who studied the application of the ESR in human medicine in 1921. Today, it is widely applied in human medicine and modified, or alternate methods to the original Westergren are currently employed. In veterinary medicine, the ESR was used in the 1940s–1960s and then progressively abandoned or rarely employed due to the amount of blood required. However, since 2020, a modified Westergren method has revived interest in ESR for dogs and cats, with several studies establishing reference intervals for these species. This review highlights the use of the ESR in dogs and cats, from the past to today, focusing on reference intervals in order to establish the physiological or pathological aspects that could affect this test.

## 1. Introduction

The erythrocyte sedimentation rate (ESR) is a biological blood phenomenon, which describes the aggregation of anticoagulated red blood cells (RBCs). Under the force of gravity, erythrocytes settle in a tube held in a vertical position, thus the plasma separates from the blood cells in the upper part of the tube. The length of blood cell-free plasma observed after a prefixed time is the ESR value [1,2].

The history of this test dates back to the 19th century when Biernacki initially discovered the ESR phenomenon in 1897. Later, Hirszfeld in 1917 and Fåhræus in 1918 independently rediscovered different applications of ESR. In 1921, Fåhræus suggested using ESR as a pregnancy test. In the same year, Westergren described this assay based on blood sedimentation observations in human patients with pulmonary tuberculosis and established the reference method for measuring the ESR [3].

This review on ESR aims to describe (a) the applications in dogs and cats from the past to the present; (b) its current possible use as a point-of-care “sickness index” thanks to the availability of a new and innovative procedure.

A comprehensive search was conducted by consulting the main web literature sites (PubMed Central, ResearchGate, National Library of Medicine, Wiley Online Library, and Google Scholar). Initially, the search focused on the ESR in dogs and cats in general, followed by an exploration of the various diseases in these animals. A possible modification of the ESR parameter was also investigated.

The following search terms were used: “ESR in dogs and cats”, “ESR and renal diseases”, “ESR and infectious diseases”, “ESR and orthopaedic diseases,” and “ESR and inflammatory pathologies”. The articles were retrieved and information regarding a possible correlation between ESR and various inflammatory diseases in both dogs and cats was extracted. Attention was paid to the reporting of reference intervals for several techniques adopted in canine and feline medicine including a new technique that has already been documented by several specialist papers. In addition, physiological, para-physiological, or pathological aspects that could affect the ESR assay were explored.

## 2. The Erythrocyte Sedimentation Rate

Three phases of the erythrocyte sedimentation process can be distinguished (Figure 1) [1,4]. The first phase (lag phase) is called aggregation, which includes the few minutes necessary for erythrocytes to adhere to each other in order to build aggregates, which are called rouleaux. The surface of the RBC is negatively charged by sialic acid molecules of the membrane proteins. RBCs, therefore, repel each other, but at the same time, the Van der Waals forces bring them together, creating an equilibrium status. In the next phase, called precipitation or sedimentation, RBCs fall more rapidly at a constant speed, and the deposition increases. In this phase, the ESR can be described by applying the modified Stokes law. This equation calculates the settling speed of a single particle such as an RBC in a fluid [5]. The Stokes law was modified to include the hindered settling effect caused by the interference of upward flow on the sedimentation speed. In the third phase, called packing, the rouleaux aggregates pile up at the bottom of the tube.

However, the ESR is a complex phenomenon influenced by various factors such as hematocrit (Hct) and plasma proteins, and specifically fibrinogens and globulins as described below [6,7].

## 3. Methods for Measuring the Erythrocyte Sedimentation Rate

The Westergren is the current reference method for measuring the ESR. It has been recommended and standardized in human medicine by the International Council for the Standardization of Hematology (ICSH) in several consecutive guidelines published over four decades [8,9,10,11,12,13]. According to this method, the blood sample drawn by venipuncture is promptly diluted with sodium citrate (ratio 4:1) and gently mixed, ensuring that no hemolysis and bubbles occur. The diluted blood is then aspirated into graduated glass pipettes. These pipettes need to stand vertically on a rack for one hour (Figure 2). During sedimentation, the pipettes should be kept at a constant temperature (18–25 °C ± 1), away from vibrations and direct light. After one hour, the sedimentation is calculated as a measure of the height of the constituted clear plasma column at the top of the pipettes. This is expressed in millimeters per hour; the ESR, thus, represents the speed of the flow [8,9,10,11,12,13].

Over the years, other manual methods have been employed to measure this parameter in human and veterinary medicine, as well as the Wintrobe method, as a comparison to the Westergren [2,14,15,16,17]. One study highlighted the superiority of the Westergren technique compared to the Wintrobe method [14,15].

In some eastern European countries, including Russia, another technique called the Panchenkov method was developed, which is more similar to the Wintrobe method than to the original Westergren method. The Panchenkov method is a micro-method that has also been applied in veterinary science [18,19,20].

In recent years, the gold standard Westergren method has been supplanted by modern devices, which can be categorized into two groups: modified Westergren methods and alternate methods [13]. The modified Westergren methods are based on the same principles as the reference method, which measures the sedimentation, but with modifications, such as a reduced reading time and the use of different types of anticoagulants. However, these modified methods enable faster measurements by using calculation algorithms that convert the value obtained in less than the one hour required by the Westergren method.

The alternate methods, on the other hand, are not based on real sedimentation, and include approaches such as centrifugation and photometric systems. Some alternate methods withdraw blood directly from the collection tube by a closed aspiration needle into a capillary tube where it is accelerated via a “stopped flow” circuit, and where the ESR is assessed. Using an infrared-ray microphotometer, the electrical pulses collected by a photodiode detector are directly correlated to the concentration of erythrocytes in the capillaries. This method is used to delineate a sedimentation value by applying a mathematical algorithm. Although they do not measure the sedimentation, the ESR results are obtained in a much shorter time than the Westergren reference method; however, the procedure consumes too many blood samples and there is considerable waste [13].

## 4. Erythrocyte Sedimentation Rate in Human Medicine

The ESR test is still the most widely performed laboratory assay to monitor inflammation in human medicine. It is a useful tool for investigating the pathophysiology of several disorders, particularly cardiovascular, hematological, neoplastic, and rheumatic diseases [21]. During an inflammatory condition, the organism produces typical plasma acute phase proteins such as fibrinogen, immunoglobulin M (IgM), and α2-macroglobulin which collectively have agglomerant properties. Agglomerins have a high affinity for glycoproteins on the RBC membrane and can act as a bridge between two cross-linking erythrocytes. They also function as neutralizers, as their positive charges nullify the negative charges of acid sialic residues on the RBC surfaces, which would otherwise repel each other. These proteins facilitate the aggregation of RBCs, and the speed of sedimentation increases [22,23]. ESR results can be reduced by erythrocytosis and by several morphological changes in RBCs, such as microcytosis, spherocytosis, or other types of poikilocytosis, while an increase can be observed in macrocytosis. In conclusion, the ESR is considered as a non-specific marker of inflammation, and it should be complemented with other hematological tests for a comprehensive health assessment [1,24,25,26]. The ESR has also been shown to increase in several other conditions where inflammation is not the primary cause of disease, such as renal disease, orthopedic disorders, etc.

The ESR in humans is influenced by several physiological factors, including age and gender. Typically, females tend to have a higher ESR compared to males. There is also an observable ESR increase with advancing age, with lower values in pediatric and higher in elderly patients. Technical variables such as high room temperature and the use of tilted tubes may contribute to a higher ESR. Pregnancy also induces an ESR increase [1,24,25,26].

Some preanalytical factors may also affect the ESR in patients. Of these, the negative correlation between hematocrit and ESR plays a major role, and thus a formula for the ESR correction based on Hct has been proposed [6,15,27,28,29,30]. However, contrasting findings have been reported in relation to this correction. In fact, the original application of the Fabry formula to ESR values slightly above the reference limit converted a pathological value into a healthy value, potentially masking an underlying alteration. Similarly, for a value-added diagnosis in the case of anemia, the ESR correction based on Hct value may be useful [6,25,27,28,29,30]. This aspect warrants further investigation.

In patients with very severe regenerative anemia, the Westergren method shows a biphasic pattern, i.e., the absence of clear separation between the settling erythrocytes and the plasma in the sample tube. This is related to the presence of immature erythrocytes, including reticulocytes or a high number of erythrocytes with abnormal shapes. Additionally, icteric samples produce a dark yellow plasma which may be difficult to differentiate from the sedimented RBCs. In human medicine, these interferences, along with the influence of clots on the ESR, are being investigated in order to establish the sample acceptance criteria to perform the ESR test [2].

In human medicine, it has been reported in the literature that certain drugs, particularly those with anti-inflammatory properties (such as non-steroidal anti-inflammatory drugs and corticosteroids) or antipyretic effects (such as acetylsalicylic acid), can lead to a reduction in ESR values. The reduction may also occur with the administration of anesthesia (propofol or thiopental) or albumin infusions, valproic acid, or statins, while other substances such as heparin may increase the ESR [1,2,24,25,31,32].

## 5. The Erythrocyte Sedimentation Rate in Veterinary Medicine

The first paper recorded on the application of ESR in veterinary medicine dates back to 1940 [33] and uses the Wintrobe technique to evaluate this parameter in dogs. The appropriate use of ESR was then established by Schalm in 1965 in his famous book, Veterinary Hematology (and reported in the following editions of his book), as mentioned in Frank’s Master of Science thesis [34]. In the same years, a few papers were published on the measurement of ESR in sheep and horses [35], and later in dogs and cats [36], adopting the Westergren or the Wintrobe method. A chart was proposed to correct the canine ESR based on the packed cell volume, which is similar to the hematocrit [36,37,38]. Also, the biphasic pattern of the ESR phenomenon has already been reported in dogs [37] (Figure 3). The test then fell into partial disuse due to the manual method that had long turn-around times (TATs), and which was laborious and required too much blood, which is critical, especially for small animal species (cats or small dogs). In the last few decades, the ESR regained some popularity, and a few papers were published mainly using the Westergren method to assess the health status of dogs [39,40,41,42].

Lastly, the MINI-PET (DIESSE Diagnostica Senese S.p.A. Monteriggioni, Italy), which is an automated instrument for ESR assessment, has been recently developed. The MINI-PET is based on the modified Westergren method using EDTA anticoagulated blood without using up the entire sample. To perform the MINI-PET ESR test, the same tube required for an automated cell blood count is introduced into the analyzer. This prevents the need for multiple animal samples. The vials are not opened, no reagents are required, and thus no additional waste products are generated. The ESR is obtained by optoelectronic sensors that measure the sedimentation of blood in 14 min for dogs and cats. Compared to the Westergren method, this ESR procedure is totally automated (apart from a gentle resuspension of blood by inverting the tubes before introducing them into the device), reduces the amount of blood needed for routine blood tests and the TAT, and ensures maximum safety for the operator.

A few studies have adopted this new device to evaluate the sedimentation in dogs, cats, and horses [43,44,45,46,47,48,49,50,51,52]. Two of these studies [43,47] describe the analytical performance of the instrument compared to the reference Westergren method. The analytical validation was carried out according to international guidelines [13] and has been published only for horses. A good correlation with the manual method was reported as well as a high level of precision (CV% < 10%) [47].

The rapid TAT can greatly improve the workflow in veterinary clinic laboratories and research settings, making the MINI-PET a reliable tool for quick ESR measurements.

## 6. Application of ESR in Dogs

### 6.1. ESR Values in Healthy Dogs

Reference intervals (RIs) are an important tool to define the range of values of an analyte or measurement and to distinguish between healthy and unhealthy patients. In veterinary medicine, the American Society for Veterinary Clinical Pathology’s (ASVCP) dedicated guidelines to improve laboratory quality and to determine de novo reference intervals, recommend the minimum numbers of enrolled animals and the appropriate statistical approach [53,54].

A few studies in the literature have attempted to define non-pathological canine ESR, with some evidence regarding the influence of breed, sex, reproductive status, and age of the population selected for the definition of RIs [33,34]. Different techniques in terms of ESR assessment have also been used. In addition, many clinicians have created their own control groups by enlisting healthy animals. Table 1 reports the studies where the ESR RIs were investigated in healthy dogs, and the other papers show only the control group. To compare the results, the ESR values are obtained, and the adopted method are also indicated.

The first investigation on the definition of RIs using the Westergren reference method was published in 2003 [42]. Other studies employed different techniques with similar results or else the methods were not always clearly reported. In Andonova [62], the RI proposed using the Panchenkov method was 0–5 mm/h which was very similar to the RI reported by Briend-Marchal [59].

The reference interval using the MINI-PET was reported in two papers that were published a few years apart. The former is a paper by Militello [43] which established an interval of 0–10 mm/h for the modified ESR obtained in 20 min. A comparative statistical study using the Westergren method reported an RI that was about half that determined for the MINI-PET (RI 0–5 mm/h). The latter was a prospective cohort study on 120 healthy dogs conducted by Gori [48] using the MINI-PET. The RI, based on the 2.5th–97.5th percentile results, was established between 1 and 8 mm/h in 14 min. The different RI compared to that reported by Militello [43] is due to the use of an updated version of the MINI-PET [48].

These data provide additional evidence that the maximum ESR values, irrespective of the method used, mostly remain below 10 mm/h. The only exceptions were in the studies by Coles [38], Yogeshpriya [68], Sharma [56], and Paltrinieri [51].

These exceptions should be further investigated to establish whether they are related to the breed and to establish more reliable RIs. They could represent a gray zone where it is difficult to assess the real pathological value.

Several of the studies reported in Table 1 focus on the relationship between ESR and age or sex. Frank [34] arranged three groups of dogs based on their age (ranging from 1 month to 6 years) and measured the ESR with the Wintrobe technique. The investigation showed that despite a small variation with age, all the groups remained below 4 mm/h, and no significant differences between the age groups were found. On the other hand, Briend-Marchal [59] obtained different results by analyzing the hemogram, fibrinogen, proteins, and ESR (using the Westergren technique). They found that the ESR increased with age, up to 20 mm/h. The authors suggested that the ESR increase in elderly dogs was related to generic cell factors such as modifications in the erythrocyte membrane with age. Khan [42] reported ESR differences related to sex (obtained with the Westergren method in one hour). Comparing the results revealed small but significant differences in male and female dogs that did not influence the upper limit of the RI.

In addition, pregnant females showed a higher ESR (3.6 ± 0.6) than non-pregnant females (3.2 ± 0.1) with an almost significant *p*-value. No statistical differences in ESR were observed in the dogs considering age or body condition score.

### 6.2. Application of ESR in Canine Infectious Diseases

ESR values increase in human infection and are significantly higher in bacterial than in viral diseases [72]. The ESR is usually evaluated with other acute phase reactants with different kinetics (e.g., C-reactive protein, CRP), particularly in the follow-up of patients with osteomyelitis, suspected prosthetic joint infection, and more generally as a prognostic marker of disease course [73].

Most of the information available on the ESR evaluation in canine infectious diseases concerns vector-borne diseases caused by some bacterial (*Anaplasma phagocytophylum*, *Ehrlichia canis*, or *Rickettsia rickettsii*) and protozoal pathogens (*Babesia* spp., *Leishmania infantum*, or *Trypanosoma* spp.) and by filarial worms (*Dirofilaria immitis*, *Dirofilaria repens*, or *Acanthocheilonema reconditum*). Scarce data on leptospirosis and sarcoptic mange are also available [57,63,74]. Table 2 and Table 3 detail the ESR numerical data available to date in the literature; the correlation with other hematological parameters, if available; the ESR technique; and the type of study, including the number of animals.

Granulocytic anaplasmosis (caused by *A. phagocytophylum*), canine monocytic ehrlichiosis (caused by *E. canis*), Rocky Mountain spotted fever (caused by *R. rickettsii*), babesiosis, trypanosomiasis, and leptospirosis can manifest as an acute febrile disease, which is potentially lethal if untreated and a systemic inflammatory response occurs. Monocytic ehrlichiosis can also follow a severe chronic course and a chronic inflammatory disease is typically caused by *L. infantum* and *Dirofilaria* spp.

Some case reports and case series have described high ESR values measured in all these conditions [40,41,45,48,49,50,51,55,58,60,61,63,66,70,71,74,75,76,77,78,79,80,81,82]. A possible ESR application has been shown in these studies, for example, an inflammatory or disease-stage marker. Interestingly, in the case of *L. infantum*, the ESR measurement also differentiated dogs with clinical leishmaniosis from dogs exposed to or infected by the parasite [45,50]. The clinical management of a single infection by *Babesia gibsoni* [81] and multiple infections with *Leptospira interrogans*, *B. gibsoni*, and *D. repens* have been reported in the literature [82]. The ESR values seem to increase in multiple infections. In fact, dogs infected by *D. immitis* and concomitant clinical leishmaniosis had significantly higher ESR values than dogs with only heartworm or *L. infantum* infections [49].

Similarly to a mirror that reflects the general inflammatory condition and the concentrations of plasma proteins in the acute phase, the ESR could be used to monitor the effectiveness of a pharmacological treatment. The return of high ESR measurements within the RI has been reported after specific treatment in dogs with babesiosis [61] and trypanosomiasis [79]. Positive correlations with other positive markers of inflammation such as fibrinogen, CRP, and total globulins, and a negative correlation with negative markers of inflammation, such as iron, albumin, and albumin–globulin ratio, have been found [40,45,48,50,51,61,80]. Two studies found that the ESR was the most sensitive index of inflammation studied in their experimental conditions [45,50].

Additional data on the kinetics of ESR in the course of canine infectious diseases and the correlation of a higher ESR with the infective dose have been found in experimental transmission studies with *E. canis* [83,84], *R. rickettsii* [85], and *Sarcoptes scabiei* [57].

**Table 2 animals-15-00246-t002:** An overview of ESR values in different types of canine bacterial infectious diseases.

Type of Study (Enrolled Dogs)	Pathogen	Main Laboratory Results (ESR Values in mm/h)	ESR Method	Ref.
Case report (n = 1)	*Leptospira icterohaemorrhagiae*	Anemia and high ESR value (80)	N.R.	[74]
Prospective controlled (n = 36 sick; n = 12 controls)	Dogs seropositive to *Leptospira* spp.	Higher ESR (34.7 ± 26.9) ^§^^	N.R.	[63]
Case report (n = 1)	*Anaplasma phagocytophilum*	Thrombocytopenia and very high ESR (83)	N.R.	[76]
Experimental infection (n = 34)	*Erlichia canis*	Thrombocytopenia followed by anemia. High ESR (50) in the acute phase; gradual normalization after treatment ^	Wintrobe	[83]
Review (n = 2)	*E. canis*	High ESR values, along with hypergammaglobulinemia, hypoalbuminemia, and pancytopenia, in a dog with a febrile peak (64) and in a dog with chronic recurrent epistaxis (42)	Wintrobe	[75]
Experimental infection (n = 14)	*E. canis*	Infection-induced CBC changes and high ESR (N.R.) in 6 dogs	Wintrobe	[84]
Observational controlled (n = 17 infected; n = 11 controls)	*E. canis*	Anemia, thrombocytopenia, lymphocytosis, monocytosis, and high ESR (58.7) and CRP ^	Westergren	[40]
Observational multicentric (n = 32)	*E. canis*	No increase in ESR (26.8 ± 23) ^§^ compared with the reference range ^	N.R.	[71]
Experimental controlled infection (n = 26 infected; n = 6 controls)	*Rickettsia rickettsii*	High ESR, especially in dogs receiving the two highest infective doses	Wintrobe	[85]

Legend: n, number; N.R., not reported; ^ range of values are reported in Table 1; ^§^, mean ± standard deviation; CBC, complete blood cell count; CRP, C-reactive protein.

**Table 3 animals-15-00246-t003:** An overview of ESR values in different types of canine parasitic infectious diseases.

Type of Study Enrolled Dogs)	Pathogen	Main Laboratory Results (ESR Values in mm/h)	ESR Method	Ref
Observational controlled (n = 50 sick; n = 20 controls)	*Babesia* spp.	Increase in CRP, SAA, HPT, and ESR ^. Positive correlation of ESR with CRP and SAA. ESR decreased after imidocarb treatment but with significantly higher values compared with healthy dogs receiving the same drug.	Westergren	[61]
Observational cases series (n = 6)	*Babesia gibsoni*	Various clinical pathological abnormalities including high ESR (31.13 ± 1.10) ^§^	N.R.	[81]
Retrospective case series (n = 21)	*Babesia* spp.	CBC abnormalities and increased ESR	N.R.	[77]
Case–control series (n = 50 infected; n = 50 controls)	*Babesia* spp.	Higher ESR in infected dogs (21.8 ± 4.3) ^§^^	N.R.	[41]
Observational controlled (n = 15 sick; n = 15 controls)	*Babesia* spp.	Higher ESR in dogs with hemoglobinuria (24.63 ± 4.25) ^#^ and oliguria/anuria (28.35 ± 5.81) ^#^^	N.R.	[70]
Observational controlled (n = 36 infected; n = 22 controls)	*Leishmania infantum*	Higher ESR in dogs with active leishmaniosis (23.1 ± 16.6) ^§^ compared to seropositive dogs (9.5 ± 3.3) ^§^ and controls *^*. ESR correlates with fibrinogen, globulin, CRP, Hb, HPT, A/G, and Hct.	MINI-PET	[45]
Observational controlled (n = 43 sick; n = 25 seropositive)	*L. infantum*	Higher ESR (39) ° in sick dogs. ESR correlates with fibrinogen, Hct, albumin, iron, A/G, and nonsegmented neutrophils.	MINI-PET	[50]
Case report (n = 1)	*Trypanosoma* spp.	High ESR (32) and low Hb and Hct	Wintrobe	[78]
Case report (n = 1)	*Trypanosoma* spp.	Anemia and high ESR (51)	N.R.	[86]
Case report (n = 1)	*Trypanosoma evansi*	Anemia and high ESR (N.R.)	N.R.	[79]
Observational controlled (n = 8 sick; n = 22 controls)	*Trypanosoma congolense*	Higher ESR (9.63 ± 3.35) ^§^^ and prolonged capillary refill time, bleeding time, clotting time, and prothrombin time.	Westergren	[66]
Observational controlled (n = 47 infected/coinfected; n = 22 controls)	*Dirofilaria immitis*, *L. infantum*	Higher ESR (12.5) ° in dogs with *D. immitis* infection (29 also seropositive for *L. infantum*) ^. Higher ESR values (46) ° in 8 dogs with dual infection and clinical leishmaniosis. ESR correlates with other inflammatory markers.	MINI-PET	[49]
Observational controlled (n = 21 infected; n = 33 controls)	*D. immitis*	Higher ESR (36.75 ± 21.22) ^§^^	Westergren	[56]
Observational controlled (n = 36 microfilaremic; n = 15 controls)	*Dirofilaria repens*	Higher ESR in dogs with mild (n = 12; 14.08 ± 2.77) ^§^, moderate (n = 12; 29.42 ± 3.46) ^§^, and severe (n = 12; 30.25 ± 6.54) ^§^ microfilaremia ^ along with other CBC changes	Wintrobe	[58]
Observational controlled (n = 10 microfilaremic; n = 5 controls)	*Acanthocheilon-ema reconditum*	Higher ESR (3.73 ± 0.16) ^§^^ along with moderate macrocytic anemia, thrombocytosis, and biochemical changes	N.R.	[60]
Experimental controlled infestation (n = 15 sick; n = 4 controls)	*Sarcoptes scabiei* var. *canis*	Progressive reduction in RBC, Hb, and Hct and increase in WBC, neutrophils, and ESR (8.5 ± 12.5) ^§^. Normalization of ESR in four weeks post-treatment (2.1 ± 5.0) ^§^.	N.R.	[57]

Legend: n, number; N.R., not reported; ^ range of values are reported in Table 1; CRP, C-reactive protein; SAA, serum amyloid A; HPT, Haptoglobin; Hb, Hemoglobin; A/G, albumin–globulin ratio; Hct, hematocrit; CBC, complete blood cell count; RBC, red blood cell; WBC, white blood cell; ^§^, mean ± standard deviation; ^#^, mean ± standard error; °, median. The paper by Aijth et al., 2016 [82] reported the clinical management of a dog affected by triple infections. Due to the journal copyright, we are not able to show more data.

### 6.3. Application of ESR in Canine Renal and Urinary Tract Disorders

In human renal disorders, the ESR has been used especially in patients affected by end-stage and chronic kidney disease, and in patients undergoing hemodialysis [87,88,89]. Inflammation is also associated with different chronic nephropathies, and ESR increases with the acute phase in human tubulointerstitial diseases. ESR was higher in patients with a higher inflammation score in nephropathy [90].

In veterinary medicine, chronic kidney disease (CKD) is one of the most frequent disorders affecting the kidneys in dogs and cats [91]. CKD is caused by the structural and/or functional impairment of one or both kidneys that has persisted for more than approximately 3 months. In some patients, CKD may be complicated by concurrent prerenal and/or postrenal problems that may worsen the condition. However, with appropriate therapeutic management, the situation can improve [92]. The diagnosis is confirmed by imaging, systemic blood pressure measurement, urinalysis, hematologic, and serum biochemical evaluations which help to define the CKD stage [92]. Cats and dogs may be affected by acute renal impairment, acute kidney injury (AKI), which may need renal replacement therapy or hemodialysis [91]. Paltrinieri [51] reported high ESR values in dogs with CKD and overt urinary disorders when systemic inflammation was a significant pathogenetic mechanism. A world apart is represented by lower urinary tract diseases which are very different from CKD and include conditions that affect the urinary bladder and urethra [93].

The first veterinary research on ESR in renal patients was focused on glomerulonephritis, which is a common feature in kidney disease accompanied by renal failure. Two old investigations dating back to around the 1960s and 1980s (Table 4) first reported an increased ESR in these nephropathic dogs. It took another forty years for new papers on this topic to be published by Sawale [94] and Ghosh [69]. The ESR was evaluated in dogs with CKD and in both papers, it appeared elevated in nephropathic dogs, and in one of the two studies, there was a significant ESR reduction after dialysis treatment [69]. All the blood parameters in fact improved, including the kidney function analytes and anemia.

Amyloidosis is an unusual cause of renal dysfunction. It represents a group of diseases all with the deposition of similar amyloid proteins in each tissue. Usually, small local amyloid depositions do not induce clinical problems. However, sometimes amyloidosis is characterized by insoluble deposits that occur in specific organs and tissues which impair their functionality and are indicative of an underlying chronic disease [95]. The clinical signs, therefore, vary and relate to the organ where the main deposit occurs, and the affected organ is sometimes species-dependent [96]. In Shar-Pei dogs, kidneys are the main target organ for amyloid deposition. In fact, an increased ESR was recently observed in Sharp-Pei dogs affected by familial amyloidosis, which, in the final stage, predominantly affected the kidney, liver, and myocardium [97].

Table 4 reports all the detailed kidney disease data with the respective ESR values.

**Table 4 animals-15-00246-t004:** An overview of ESR values in several canine kidney diseases.

Type of Study (Enrolled Dogs)	Diagnosis	Main Laboratory Results (ESR Values in mm/h)	ESR Method	Ref
Prospective experimental (n = 22)	Acute and chronic nephritis	Anemia and high ESR (N.R.) generally one week after the experimental induction or renal damage. Progressive decrease in the following weeks	Westergren	[98]
Retrospective (n = 79 sick, n = 21 healthy)	Pyometra-associated nephritis	Higher ESR (28.2 ± 33.2 ^§^) ^. Reduction in ESR (24.8 ± 20.8) ^§^ in 30 bitches after surgery	N.R.	[55]
Retrospective (n = 44)	Renal amyloidosis	High ESR in 30/32 dogs (N.R.), some dogs with anemia	N.R.	[99]
Retrospective (n = 20)	Renal amyloidosis and thrombosis	High ESR (N.R.) in 12/20 dogs	N.R.	[100]
Case report (n = 1)	End stage CKD	Azotemia, anemia, and high ESR (45)	N.R.	[94]
Case report (n = 1)	CKD	Anemia and high ESR (32) before the hemodialysis in comparison to two post-treatment (18)	N.R.	[69]
Prospective (n = 25)	CKD and urinary disorders	High ESR values (N.R.)	MINI-PET	[51]

Legend: n, number; N.R., not reported; ^, range of values are reported in Table 1; CKD, chronic kidney disease; ^§^, mean ± standard deviation.

### 6.4. Application of ESR in Canine Orthopedic Disorders

The most common and important application of the ESR test in human medicine is in the diagnosis and assessment of rheumatoid arthritis (RA) [101,102]. The “28-joint Disease Activity Score” incorporates the ESR (DAS28-ESR) with the most common 28 tender swollen joints affected in this disease. It represents a key parameter of chronic inflammation and clinical decisions in people with RA [103]. Although the ESR is not considered a test for the acute phase, it is also used in some orthopedic contexts due to its differences in the half-life compared to CRP [104]. The International Consensus Group on Fracture-related infections (FRIs) includes both ESR and CRP in the protocol for FRIs which are the most frequent complications following fracture fixation [105].

In dogs, the ESR has been assessed in several orthopedic disorders (Table 5) such as inflammatory arthritis, RA, and osteoarthritis. It was found to be higher than the RI of dogs with non-inflammatory arthropathies [106,107]. In addition, both longitudinal studies and review articles have reported that the ESR could be a valuable diagnostic or prognostic tool, along with other hematological or biochemical markers of inflammation in canine osteoarthritis [39,64,108].

Other studies have evaluated the potential role of the ESR in monitoring the follow-up of dogs subjected to different treatment protocols for fracture repair. These studies indicated that the use of ESR may help to identify protocols characterized by less intense post-surgical inflammation, as detailed in Table 5 [109,110].

**Table 5 animals-15-00246-t005:** An overview of ESR values in canine orthopedic disorders.

Type of Study (Enrolled Dogs)	Diagnosis	Main Laboratory Results (ESR Values in mm/h)	ESR Method	Ref
Observational (n = 4)	Arthritis and degenerative joint disease	High ESR in one dog with arthritis (41); lower ESR (2–5) in 3 dogs with a degenerative joint disease	N.R.	[106]
Observational (n = 30)	Rheumatoid arthritis	High ESR (N.R.) in 24/30 cases	N.R.	[107]
Experimental (n = 43 sick; n = 30 controls)	Osteoarthritis	High ESR (N.R.) at 30, 60, and 120 min	N.R.	[108]
Review (n = N.R.)	Osteoarthritis	ESR is a useful diagnostic test, quick and simple to detect the presence of inflammation	Westergren	[64]
Observational longitudinal (n = 10)	Knee osteoarthritis	Increased ESR (N.R.) over time as a prognostic marker	Wintrobe	[39]
Observational Longitudinal (n = 30)	Fracture repair with antibiotics (Gr I); without antibiotics (Gr II)	High ESR in 24 dogs before surgery (Gr I = 8.05 ± 0.30; Gr II = 8.33 ± 0.56); 7 days post-surgery (Gr I = 8.26 ± 0.17; Gr II = 10.13 ± 0.73); up to 60 days post-surgery (Gr I = 4.5 ± 0.23 to 3.17 ± 0.35; Gr II = 6.67 ± 0.41 to 3.57 ± 0.56)	Wintrobe	[109]
Observational longitudinal (n = 6 experimental Gr I, n = 6 experimental Gr II, and n = 6 controls Gr III)	Fractures of tubular bones of various etiology	Higher ESR ^^ 10 days post-surgery (Gr I = 14.67, Gr II = 10.67, Gr III = 11.17). Progressive decrease at 20 (Gr I = 12.50, Gr II = 6.67, and Gr III = 12.17) and 30 days post-surgery (Gr I = 67, Gr II = 3.50, and Gr III = 7.33)	Panchenkov	[110]

Legend: n, number; N.R., not reported; ^^ ESR units in the paper reported as mm/s but should be mm/h; Gr, group.

### 6.5. Application of ESR in Miscellaneous Disorders

As shown in Table 6, the prognostic role of ESR was assessed in different surgical protocols for splenectomy. However, no differences were recorded probably due, irrespective of the protocol, to persistent anemia and leukocytosis [111]. In addition, a comparison of the ESR kinetics studied over time in dogs with different inflammatory disorders evidenced high ESR values in dogs with anemia, fever, abscesses [112], urinary tract infections [68], and experimental infections with different bacteria [62,113]. More generally, the same trend was observed in inflammatory disorders characterized by anemia or leukocytosis or increased acute phase proteins [50], and also in diseases in which the inflammation was probably secondary to another primary disease such as CKD or neoplasia [51].

## 7. Application of ESR in Cats

The ESR evaluation in feline medicine has been less documented in the literature because the amount of blood collected to be used only for ESR was rarely carried out to include the Westergren or Wintrobe method. In fact, the need to preserve blood for other laboratory exams is more important than its use for just one test. Despite the low amount of blood routinely collected from cats, tests requiring no blood at all or at least in very low quantities should be encouraged rather than other tests that use higher amounts of serum to measure inflammatory markers.

### 7.1. ESR Values in Healthy Cats

Similarly to dogs, reference feline intervals have also been defined according to the ASVCP guidelines. In some papers, healthy cats have been used as control groups, but only in very few others to establish the reference interval for the diagnosis and to monitor health status (Table 7).

An ESR reference interval of 0–5 mm/h, presumably obtained with the Westergren method, was defined by Farrow [114]. In contrast, other recent studies have proposed a wider feline RI. In the study by Uva [46], the ESR values of cats with CKD were compared to a control group of 10 healthy cats and showed a median value of 30 mm/h, (range 16–37 mm/h). Donato [52] used a control group of 10 healthy cats and showed a median value of 13.5 mm/h, with 9.8–24 values as the 25th and 75th percentiles.

Chekrysheva and Rodin [115] reported that the ESR after treatment for feline mastitis was 7.8 ± 1.89 mm/h. Additionally, in a clinically healthy group of 11 cats used as a control to study cholangiohepatitis, the average ESR was 4.3 ± 0.52 mm/h [116]. Lastly, another investigation of the feline ESR RI on 52 healthy cats proposed a minimum and maximum range of 1–23 mm/h [117].

### 7.2. Application of ESR in Feline Diseases

CKD is one of the most frequent inflammatory diseases in cats, and currently the International Renal Interest Society (IRIS) classification is used to stage it [118]. The first scientific article on the ESR in feline kidney disease is a case report of a cat with nephrotic syndrome and a very high ESR compared to the control group [114]. These results were recently confirmed by Uva [46] in a study on a larger number of cats that associated more advanced stages of kidney disease with higher ESR values.

An increase in ESR was also found in cats affected by viral infections, i.e., FeLV (feline leukemia virus) [119], calicivirosis [120], and FIV (feline immunodeficiency virus) with or without *L. infantum* antibody-positivity [52]. In addition, an ESR increase was also found in cats with purulent mastitis [115], acute bacterial cholangiohepatitis [116], and in cats affected by several diseases where the ESR correlated positively with fibrinogen [117] or SAA [46,117]. Donato et al. [52] found elevated ESR values in cats with antibody positivity to *Leishmania,* FIV, or both. In addition, ESR positively correlated with total proteins, immunoglobulins G and M, and alpha-1-globulins, and negatively with total iron binding capacity, albumin, and the albumin–globulin ratio (Table 7).

**Table 7 animals-15-00246-t007:** An overview of ESR values in feline miscellaneous disorders.

Type of Study (Enrolled Cats)	Diagnosis	Main Laboratory Results (ESR Values in mm/h)	ESR Method	Ref
Case report (n = 1)	Membranous glomerulonephritis	Very high ESR value (74)	N.R.	[114]
Prospective experimental (n = 32 sick; N = 10 controls)	CKD	ESR similar in IRIS stage 2 cats (35) ° and in controls (30) °; high ESR in IRIS stage 3 (64) ° and stage 4 (56) ° cats	MINI-PET	[46]
Prospective (n = 7)	FeLV infection	High ESR in 5 infected cats (range 23–53)	N.R.	[119]
Prospective (n = 30)	FCV infection	High ESR (15.77 ± 0.22) ^§^ in 27/30 FCV-positive cats	N.R.	[120]
Prospective (3 treatment groups of 10 cats)	Purulent mastitis	Higher ESR values before treatment in each group (20.6 ± 2.1 vs. 7.8 ± 1.89; 19.3 ± 2.11 vs. 3.8 ± 1.15; 21 ± 1.9 vs. 3.4 ± 1.4) ^§^	N.R.	[115]
Prospective (n = 12 sick; n = 11 controls)	Acute bacterial cholangiohepatitis	High ESR in 2 treatment groups (17.8 ± 3.50; 23.4 ± 3.90) ^§^. Decrease after treatment (9.0 ± 1.29; 6.6 ± 1.02, respectively) ^§^	N.R.	[116]
Prospective (n = 35 sick; n = 10 controls)	FIV and/or *Leishmania infantum* infections	Higher ESR in *Leishmania* (29.5, 10–61) ^ç^, FIV (55.5, 13–66) ^ç^, and coinfected (45, 15–71) ^ç^ cats	MINI-PET	[52]
Prospective cohort (n = 143 sick; n = 57 controls)	Various diseases	Higher ESR (29) ° in sick cats. ESR positively correlated with fibrinogen. The highest ESR (47) ° in acute-on-chronic diseases compared with acute (16) ° and chronic (14) °	MINI-PET	[117]
Prospective (n = 30 sick; n = 6 controls)	Intestinal dysbiosis	High ESR in compensated (7.13 ± 0.52) ^§^, subcompensated (15.62 ± 0.88) ^§^, and decompensated dybiosis (24.26 ± 1.71) ^§^	N.R.	[121]

Legend: n, number; N.R., not reported; IRIS, International Renal Interest Society; CKD, chronic kidney disease; FeLV, feline leukemia virus; FCV, feline calicivirus; ^§^, mean ± standard deviation; FIV, feline immunodeficiency virus; ^ç^, median, minimum–maximum; °, median.

## 8. Conclusions and Future Directions

ESR is an old test for evaluating the inflammatory reaction. It was used in veterinary medicine alongside human medicine dating back to the 1940s–60s, when the availability of laboratory diagnostic tests was quite low, and the time required was reasonable. Despite its global use in human medicine, particularly in low-income countries, the ESR has been forgotten in veterinary medicine for many years [21,43,122]. The main reason for its abandonment has been the amount of blood required and the meticulous attention to the appropriate manual preparation of the special equipment and sample required by the Westergren or the Wintrobe methods. Other limitations include the long TAT, the requirement for a different anticoagulant from the EDTA, and the waste production.

Diagnostic devices for ESR measurements that use modified or alternate Westergren methods could increase the point-of-care use of this test. These automated analyzers allow for blood savings as they use the same EDTA blood tube to carry out routine hematology. This is of utmost importance in small patients (puppies and kittens, cats, and toy breed dogs). The application of EDTA as an anticoagulant also improves blood cell stability, enabling the formation of rouleaux and eliminating unphysiological effects on cells, which are crucial in the ESR reaction [13,123].

This literature review includes all the studies published on ESR, including a few case reports and case series. However, the majority of the studies cited in this review are large and structured (prospective, retrospective, or experimental). The type of study and the number of animals investigated are reported in the first column of the tables. Therefore, the quality of these manuscripts and the level of scientific evidence are sufficiently high to highlight the potential of ESR in canine and feline medicine.

Ultimately, these studies reveal that in dogs and cats, the ESR increases in many diseases, ranging from severe to mild, acute or chronic inflammation, infections, diseases affecting the kidney or the urinary tract, or orthopedical disorders, as well as in other conditions. Most of the studies report an increased ESR in dogs, while only a few studies are currently available on cats. The introduction of automated analyzers will improve the knowledge of ESR in cats and small-sized patients in general.

As in human medicine, many physiological and para-physiological factors can affect the ESR value [2,25,26,124]. A few papers with results from clinically healthy animals have also examined these variables, which should also be investigated more thoroughly in dogs and cats to obtain the most appropriate RI to compare with the results of patients affected by different diseases.

Another issue to be explored in future studies is the possible association between the magnitude of ESR increases and the severity of the disease. This will help to define the clinical decision limits. As shown in several of the cited studies, a very high ESR could suggest a severe or advanced stage of disease [45,46,49]. In addition, values just over the RI and without any physical signs of disease could suggest the need for further monitoring of the patient.

Finally, the influence of pre-analytical factors on the ESR needs further exploration such as severe hemolysis, lipemia, or icterus, which could affect both manual methods and automated analyzers.

As expected, this review has shown that an increased ESR occurs in a wide range of diseases, and it behaves like a generic “sickness index” as with others.

Today, the most popular markers of inflammation are acute phase proteins (CRP in dogs and SAA in cats) which have been demonstrated to be accurate in confirming clinical inflammation or in detecting subclinical inflammation [125]. These analytes can be measured with accurate methodologies and are often included in diagnostic panels offered by many laboratories. In terms of CRP, and to a lesser extent SAA, they can also be accurately determined using in-clinic assays with instruments already available in many veterinary clinics relatively cheaply and rapidly. However, the study results demonstrate that the ESR and acute phase proteins may be complementary. These markers correlate in many clinical settings [126]. Additionally, some studies have reported a different increase in these two markers linked to different disease stages [45,46,49], or a different behavior in chronic conditions or during the follow-up of acute diseases, when increases in CRP are less severe than in the acute phase, while the ESR remains high while inflammation persists. This characteristic could be useful in veterinary applications when an owner brings a pet that already has a disease to the veterinary clinic.

The new ESR automated analyzers that use the same EDTA tube for the CBC without consuming blood enable the blood to be saved for further analyses, do not require additional reagents, and are, therefore, even cheaper than acute phase protein measurements. They also have a short turnaround time, which means that information can be obtained on the possible presence of inflammation when an in-clinic measurement of CRP or SAA is not possible.

The availability of more data on ESR changes in dog and cat diseases is of particular interest. In fact, the ESR is currently the only point-of-care and low-cost test that can provide information on the inflammatory reaction.

In conclusion, although the ESR assay is old and non-specific, it could be useful in veterinary medicine. The introduction of automated systems with a short turnaround time and a low requirement for blood samples would make the test easier to use. Undoubtedly, the ESR increases nonspecifically in many pathological conditions, thus limiting its diagnostic power. However, high ESR values represent a generic warning sign for practitioners who could use it as a “sickness index” for patients who need further investigations. Additionally, studies should be conducted to demonstrate the value and application of ESR as a disease prognostic marker, as well as in monitoring drug treatments, where it could be used as a coadjutant.

## Figures and Tables

**Figure 1 animals-15-00246-f001:**
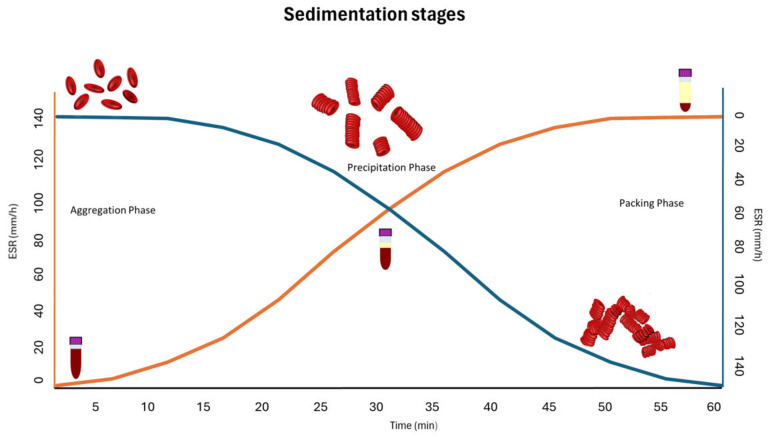
The ESR phases show the erythrocyte behavior (green-blue line and right ordinate) and the related consequence of increasing the amount of plasma (in yellow) in the sample tube (orange line and left ordinate), which are both time-dependent.

**Figure 2 animals-15-00246-f002:**
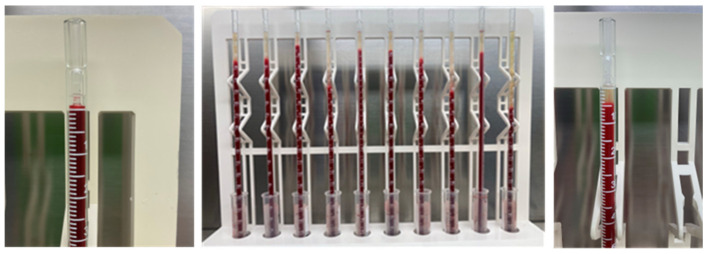
Reference manual Westergren method. In the photo on the left, the glass pipettes are filled with blood up to the top of the graduated scale and placed vertically in the rack. In the center photo, after one hour of sedimentation, the RBCs in the pipette fall to the bottom, and the plasma appears at the top of the column. The photo on the right shows a magnification of the height of the plasma column on the graduated scale which corresponds to the ESR expressed in mm/h.

**Figure 3 animals-15-00246-f003:**
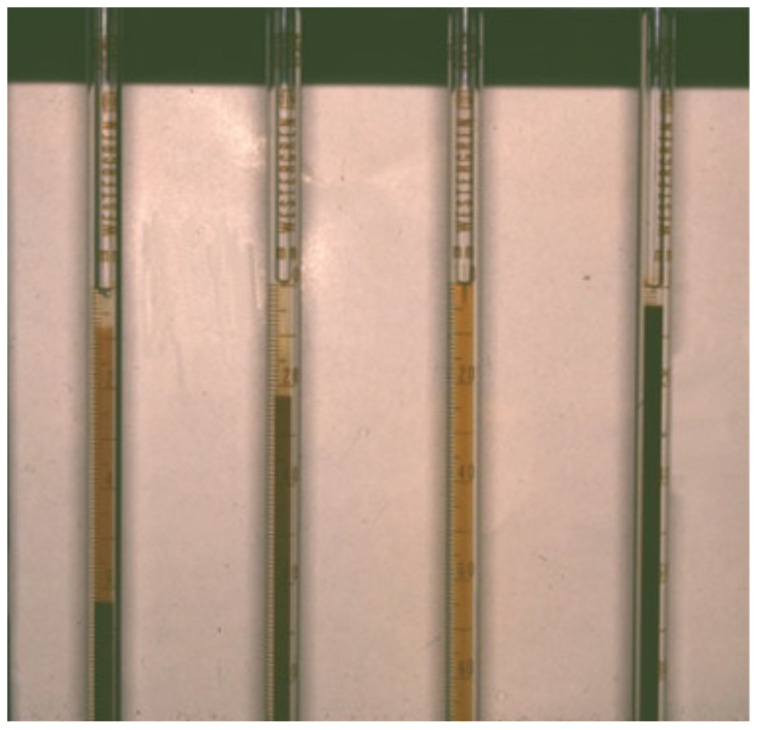
A few examples of the ESR in dogs according to the original Westergren method (kindly supplied by G. Lubas, 1997). Legend: from left to right—1st pipette, diphasic ESR with no definite separation between plasma and settled RBCs due to the cloudy red-dish area remaining between them; 2nd pipette, pathologic ESR with extended area of plasma on the top; 3rd pipette, very high ESR in icteric plasma; 4th pipette, ESR within the RI with reduced area of clear plasma at the top.

**Table 1 animals-15-00246-t001:** ESR value (mm/h) measured in healthy dogs used as a control group or for the definition of reference interval (RI) with different techniques and in different studies (listed in order of publication year).

Technique/ Method Adopted	N Dogs	ESR Values (mm/h)	Reference	Type of Group for Healthy Subjects
N.R.	21	Mn 1.7 ± SD 2.0 IR 0–8	[55]	Control group
Wintrobe	11	IR 0.5–3	[34]	Control group
Westergren	33	Mn 12.5 ± SE 5.87	[56]	Control group
Wintrobe	N.R.	IR 5–25	[38]	Control group
N.R.	4	Mn 2.5 ± SD 5.0	[57]	Control group
Wintrobe	15	Mn 0.27 ± SE 0.12	[58]	Control group
Westergren	27	<3 ^(1)^	[59]	Used for RI
N.R.	5	Mn 2.45 ± SD 0.12	[60]	Control group
N.R.	N.R.	<10 ^(2)^	[61]	Control group
Panchenkov ^(2)^	N.R.	IR 0–5	[62]	Used for RI
N.R.	50	Mn 3.4 ± SD 0.3	[42]	Used for RI
N.R.	12	Mn 3.00 ± SD 0.37 ^(3)^	[63]	Control group
Westergren	NR	IR 0–5	[64]	Control group
Westergren	10	Mn 6.00 ± SE 0.47	[65]	Control group
Westergren	22	Mn 3.85 ± SD 0.46	[66]	Control group
Panchenkov ^(3)^	25	Mn 3.7 ± SD 0.3 ^(3)^	[67]	Control group
N.R.	32	Mn 13.64 ± IR 9.2 ^(3)^	[68]	Control group
MINI-PET ^(4)^	119	IR 0–10	[43]	Used for RI
Westergren	119	IR 0–5	[43]	Used for RI
N.R.	50	Mn 3.80 ± SE 0.30 ^(5)^	[41]	Control group
Westergren	11	Mn 6.45 ± SE 2.79	[40]	Control group
MINI-PET	120	IR 1–8	[48]	Used for RI
N.R.	N.R.	IR 6–10	[69]	Used for RI
MINI-PET	22	Mn 9.09 ± SD 7.97	[45]	Control group
N.R.	15	Mn 5.13 ± SE 0.32	[70]	Control group
N.R.	N.R.	IR 5–8	[71]	Used for RI
MINI-PET	22	Md 7.5–IQR 5–11	[49]	Control group
MINI-PET	53	Md 7–IQR 2.7–10	[51]	Control group
MINI-PET	53	IR 1.0–13.7	[51]	Used for RI

Legend: N.R., not reported (the paper does not clearly state which method or technique was used for the ESR); N, number; Mn, mean; SD, standard deviation; IR, interval range; SE, standard error; Md, median; IQR, interquartile range. Notes: ^(1)^ dogs aged 1–8 years; ^(2)^ Data results derived from the graph; ^(3)^ Statistical parameter not clearly stated, and then inferred, and reported as Mn and SD; ^(4)^ Research Use Only (RUO) software 0.000; ^(5)^ the SE was inferred.

**Table 6 animals-15-00246-t006:** An overview of ESR values in canine miscellaneous disorders.

Type of Study (Enrolled Dogs)	Diagnosis	Main Laboratory Results (ESR Values in mm/h)	ESR Method	Ref
Observational (n = 12)	Total and partial splenectomy	High ESR after partial (29.7–40.5) ** or total splenectomy (27.5–32.5) **	N.R.	[111]
Observational (n = 23)	Different inflammatory conditions	High ESR associated with anemia (37) °, fever (45) °, and abscesses (35) °	Wintrobe	[112]
Observational (n = 32 sick; n = 32 controls)	Urinary tract infections	Higher ESR (25.83 ± 26.3) ^§^	N.R.	[68]
Observational longitudinal (n = 9)	Experimental infection with *Staphylococcus aureus*	Higher ESR over time: T0 (4 ± 2.2) ^§^; T48h (13 ± 11.3) ^§^; T72h (15 ± 15) ^§^	Panchenkov	[62]
Observational longitudinal (n = 5 infected; n = 5 controls)	Experimental infection with *Pseudomonas aeruginosa*	Higher ESR over time (from approximately 18 to 46 at 72h p.i.)	Panchenkov	[113]
Obsevational (n = 43 active leishmaniosis; n = 25 *L. infantum* seropositive; n = 65 inflammatory diseases)	Inflammatory diseases vs. leishmaniosis	Higher ESR in dogs with inflammatory diseases (41) ° vs. dogs *L. infantum* seropositive (11) °	MINI-PET	[50]
Observational (n = 217 sick; n = 53 controls)	Inflammatory or non-inflammatory diseases	High ESR in severe/acute diseases (5.2–26.8) **; acute/subacute inflammation (9–38) **; tumors (10–23.7) **; mild chronic disorders (5.4–11) **	MINI-PET	[51]

Legend: n, number; N.R., not reported; p.i., postinfection; **, minimum–maximum values; ^§^, mean ± standard deviation; °, median.

## Data Availability

Not applicable.

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
