# Peer review of "The Erythrocyte Sedimentation Rate (ESR) in Veterinary Medicine: A Focused Review in Dogs and Cats"

_animals, 2025, doi:10.3390/ani15020246_

Round 1
Reviewer 1 Report
Comments and Suggestions for Authors
This review about eritrosedimentation in cat and dogs is very interesting and complete. The author revise and analyses the existing knowledge on a widely used technique of great importance in animal health; however, there are no bibliographic reviews on it such as the one presented in this work.It includes an exhaustive search and a detailed analysis of the use of erythrocyte sedimentation rate detection in canine and feline clinical practice. The search includes a bibliographical analysis (it contains 119 references) and includes both classic and pioneering articles on the subject as well as more recent works. The analysis is in-depth and includes both the effects of variations in the technique and the influence of the species, age and sex.
A complete historical introduction is made, including the emergence of the different methods. After a brief description of the application of the technique in humans, a detailed description of the results of the bibliographic search in different dog and cat diseases is made. This is accompanied by tables that summarize the main results. The analysis highlights the main uses.
The division into sections is very well thought out, both in terms of which sections were used and how they were ordered, and it makes reading and searching for specific topics within the text easier. The figures illustrate some specific aspects clearly.
As a negative aspect, I think that the limitations of the technique should be discussed in greater depth.
Minor aspects
Figure 3 legend is very long
Reviewer 2 Report
Comments and Suggestions for Authors
This paper is a reasonable review of the ESR in veterinary medicine. However, is has been extensively questioned in recent years if there is any utilty ot the ESR in human medicine, and most of the veterinary papers discussing it are very old. In a world where CRP, SAA, and other acute phase proteins are readily assayed, it is unclear that there is utility to ESR. More time must be spent explaining why this test is needed compared to others with more data on them in dogs and cats.
In addition, a lot of time is spent discussing previous methods of determining ESR< which are largely deprecated now that automated analysis is possible.
Many of the tables are excessively long and need to be made much more concise
Throughout the manuscript
Comments on the Quality of English LanguageIt is crucial that it be clear, concise, and readable. The title of this paper uses improper English (it should be "focused" review). This continues to the first sentence of the paper, which is confusing and contains very poor grammar. Significant English language editing is required for clarity and readability before it is feasible to do a line by line review.
Reviewer 3 Report
Comments and Suggestions for Authors
The review addresses a relevant and underexplored area in veterinary medicine, contributing to the understanding of ESR and its utility in diagnosing inflammation in dogs and cats. The article adequately covers the historical background of ESR, measurement methods, and its application in various veterinary diseases, providing a solid foundation for future research. The inclusion of new automated methods such as the MINI-PET and comparisons with traditional methods adds value to the article, highlighting the technological evolution in ESR measurement.
- The description of the novel procedure that has revived interest in ESR in veterinary medicine is limited to this sentence.“Finally, in recent years an automated instrument for ESR assessment, the MINI-PET 211 (DIESSE Diagnostica Senese S.p.A. Monteriggioni, Italy), has been developed. It is based 212 on modified Westergren method, employing EDTA anticoagulated blood without sample 213 consuming.”As one of the pillars of the review, the MINI-PET procedure should, if possible, be described with the same level of detail as the Westergren procedure in human medicine.
- The conflict of interest paragraph should describe the relationship between DIESSE and the MINI-PET.
Comments on the Quality of English LanguageThe quality of the English language in this manuscript is generally good, with clear and precise communication of complex scientific concepts. Nevertheless, there are certain areas that could benefit from enhancements to enhance readability and fluency. In particular, some sentences are unnecessarily lengthy and complex, which can impede comprehension. Simplifying sentence structures and reducing the use of redundant connectors would facilitate the text's flow. Furthermore, minor grammatical errors, such as issues with subject-verb agreement and preposition usage, should be addressed.
(49-51). The erythrocyte sedimentation rate (ESR) is a blood biological phenomenon even now partly understood, that depicts the rate of anticoagulated red blood cells (RBCs) to aggregate and sediment in a pre-fixed time under gravity force, into a tube maintained in vertical position.
(83-85). However, the erythrocyte sedimentation is a complex phenomenon that is influenced not only by the hematocrit (Hct) but also by certain plasma proteins specifically fibrinogen and globulins as described below.
(131-132). The electrical pulses collected by a photodiode detector are directly correlated to the concentration of erythrocytes present at the capillary level and are used to delineate a sedimentation value by a mathematical algorithm.
Reviewer 4 Report
Comments and Suggestions for Authors
I think the manuscript is well written.
I have a minor question.
L25 and L34: Why is it different about 1940-1960s and 40-70s.
Please unify the words.
Round 2
Reviewer 2 Report
Comments and Suggestions for Authors
The authors have not made many of the changes requested to an adequate degree.
There is a lot of extraneous information in this paper. The tablets must be made more breif, and summary results provided. It is not clear what information is to be gained from the information as it is provided. Find the relevant data, and present it in an interpretable fasihon. Simply reducing the amount of text per line is not sufficient. This applied to the text of this paper as a whole. There is extensive repetition of the same information in different ways, and this needs to be shorter and more concise. At the same time, a lot of topics are mentioned and then not expanded on. For example, the purpose of this is described as using it as a "sickness index" and then what this means is not expanded on at all.
Ultimately, this feels like the point of this review is to try to add some marketing information for the product the authors have developed. It needs to be rewritten as a clear, focused recview of the evidence, addressing the QUALITY and not just results of papers, ina traditional review fashion
In response to some of the author's comments:
-The idea that CRP requires extensive equipment that the ESR does not is frankly incorrect. There are many point of care anlayzers and reference labs running CRP routinely. It requires analyzers that exist in many clinics worldwide. In Europe and Asia it is extermely commonly run. Less so in North Amercia, but it is still a readily available test. For most clinics, it would be easier to add a CRP to a send out panel or in hosue analyzer than to pruchase the tubes for an ESR. IF this is intended for use in limited-resource settings where send out labs and in house analyzers do not eixst, that needs to be clearly specified
There are still multiple unclear sentances. For example, in line one: "Originating the plasma part" does not make sense.
Comments on the Quality of English Language
Moderate editing still needed
